# Factors influencing the delivery and uptake of early infant diagnosis of HIV services in Greater Accra, Ghana: A qualitative study

**Antoinette Kailey Ankrah**[ORCID]*, **Phyllis Dako-Gyeke**

Department of Social and Behavioural Sciences, School of Public Health, College of Health Sciences, University of Ghana, Accra, Ghana

* ankrahkailey@gmail.com

**Data Availability Statement:** All relevant data are within the manuscript and its Supporting Information files.

**Funding:** This study did not receive funding from any source. It was self funded by the lead author

## Abstract

### Background

Early Infant Diagnosis (EID) of HIV and timely initiation of Antiretroviral Therapy (ART) can significantly reduce morbidity and mortality of HIV infected infants. Despite the benefits of early infant testing, the coverage of EID of HIV services is still low in Sub-Saharan Africa, including Ghana.

### Objectives

To ascertain the factors that facilitate or hinder the delivery and uptake of EID of HIV services.

### Methods

The study is a cross-sectional exploratory qualitative research conducted in two health facilities in the Greater Accra Region of Ghana. Respondents (n = 50) comprising health workers (n = 20) and HIV positive mothers (n = 30) were purposively sampled and engaged in in-depth interviews. The Nvivo 11 software and the Braun and Clarke's stages of thematic analysis were used in coding data and data analysis respectively.

### Results

The study found that health system factors such as inadequate Staff with sample collection skills, unavailability of vehicles to convey samples to the reference laboratory for analysis, the long turnaround time for receipt of Polymerase Chain Reaction (PCR) results, inadequate and frequent breakdown of PCR machine hindered EID service delivery. On the other hand, adequate knowledge of health workers on EID, availability of Dried Blood Spot (DBS) cards and the adoption of task shifting strategies facilitated EID service delivery. Factors such as the denial of HIV status, non-completion of the EID process due to frustrations encountered whiles accessing service and delay in receipt of PCR results served as barriers to mother's utilisation of EID services for their exposed infants. The study also identified that adequate knowledge of EID, perceived importance of EID, financial stability as well as financial support from others and the positive attitudes of health workers facilitated HIV positive mother's uptake of EID services for their exposed infants.

and was part of the lead author's research, leading to the award of an MPH degree.

**Competing interests:** The authors have declared that no competing interests exist.

## Conclusion

The factors attributing to the low coverage of EID of HIV services must be promptly addressed to improve service delivery and uptake.

## Introduction

Early initiation of HIV infected infants on Antiretroviral Therapy (ART) reduces infant morbidity and mortality by 75% and 76%, respectively [1,2]. Initiating HIV infected infants on ART can only be possible after their HIV status has been ascertained [3]. For this reason, the WHO recommends that all exposed infants are virologically tested within 4 to 6 weeks after birth [4]. This is referred to as Early Infant Diagnosis [5].

To ensure that all HIV exposed infants are diagnosed and linked to treatment if found to be infected, an EID cascade ought to be completed. This EID cascade entails a number of sequential steps: identifying HIV exposed infants; conducting PCR test on exposed infants at 4 to 6 weeks; transporting collected specimen to laboratory for processing; returning test results to both health facility and mother or guardian; and initiating ART on infants identified as HIV infected [6]. It is important that exposed infants who initially tested negative are re-tested 6 weeks after cessation of breastfeeding to ascertain if HIV infection have been transmitted postnatally via breastfeeding. Finally, an anti-body test must be conducted on all exposed infant at age 18 months [7]. It is critical that no step in this cascade is overlooked.

Irrespective of the importance of early testing of HIV exposed infants, EID coverage remains low, particularly in sub-Saharan Africa [8]. In 2015, only 54% of HIV exposed infants in the 21 Global Plan priority countries accessed EID within the first two months of life [9]. The situation is no different in Ghana, with only 3 out of every tenth child of HIV exposed infants receiving a virological test in 2016, showing a discouraging EID coverage in the country [10]. These statistics indicate that EID coverage is below the 80% level recommended by WHO [11]. Aside the smaller number of exposed infants tested, an even smaller percent receive their results or have access to ART [6,12]. Findings from previous studies have attributed cyclical stock of Dried Blood Sample (DBS) cards, delays in sample transport, limited Polymerase Chain Reaction (PCR) machine, breakdown of PCR machines, inadequate knowledge of health workers on EID, limited Staff trained with DBS sample collection skills and long turn-around time of PCR results as challenges to EID service delivery in most sub-Saharan settings [12–14]. Knowledge of EID, perceived importance of EID, financial constraints, distance to health facility, geographical relocation, religious beliefs, and attitude of health workers have been reported by previous studies as some factors that influence the utilisation of EID services by mothers [15,16].

Despite the low coverage of EID services among HIV exposed infants in Ghana, very few studies carried out in Ghana have looked at the factors contributing to the low uptake of EID services. This study, therefore, sought to ascertain the factors that facilitate or hinder the delivery and uptake of EID of HIV services.

## Materials and methods

### Study site

The study was conducted in two different health facilities, both situated in the Greater Accra Region of Ghana. This is a region noted to have high HIV prevalence in the country. Facility A

is a regional hospital located within the Accra Metropolitan area with more resources and more qualified Staff. In contrast, Facility B is a district hospital in Ledzokuku Municpal area with fewer resources and less qualified Staff.

The Greater Accra Region recorded an HIV prevalence of 2.4% in 2016 among the 15–49 age group per the HIV Sentinel Survey (HSS) Report, showing a decline from 3.2% in 2015 [17].

The Region has a number of health facilities with 274 Prevention of Mother-to-Child Transmission (PMTCT) Centres all serving as entry points for testing HIV exposed infants [17], of which the Greater Accra Regional Hospital and the LEKMA Hospital are included.

## Study design

The study employed a cross-sectional exploratory qualitative approach to have an in-depth understanding of the perspectives and experiences of health workers and HIV positive mothers providing and utilising EID of HIV services, respectively.

## Study participants

The study utilised two sets of respondents. These were health workers and HIV positive mothers. A total of 50 participants comprising 20 health workers and 30 HIV positive mothers participated in the study.

## Sampling

A purposive sampling technique was employed in selecting study participants. Criterion purposive sampling technique was further adapted in electing eligible mothers and health workers for the study. The Matron at the ART Centre assisted with the recruitment of eligible mothers for the study. HIV positive mothers who utilised the ART Centre during the data collection period, satisfied the inclusion criteria and were willing to participate in the study were selected for the interview. To ensure participating health workers were experienced, Supervisors at the antenatal, post-natal and ART units assisted with the selection of health workers. Permanently employed health workers who were directly involved in the provision of EID services for more than a year were selected. Eligible health workers who were available and willing to participate in the study were interviewed. Study participants were engaged until no new information was obtained, signifying a saturation point [18]. Saturation point was reached with periodic debriefing among the research team.

**Inclusion criteria.** HIV positive mothers who accessed services at the ART Centre and satisfied the following criteria were eligible for selection: enrolled in PMTCT sessions during pregnancy; delivered at the selected facilities; and exposed infant is at most a year old.

Health workers within the selected facilities were eligible for selection if they satisfied the following criteria: permanently employed; directly involved in the delivery of EID services; and provided EID services for at most a year prior to data collection.

## Data collection

The qualitative data collection method used for this study was individual interviews, specifically in-depth interviews (IDIs) to explore the perspectives of respondents on a phenomenon of interest [19]. Two separate interview guides were utilised for health workers and HIV positive mothers. The interview guide for the health workers discussed health system factors influencing the delivery of EID services as well as relevant recommendations. That of the HIV Positive mothers inquired about their knowledge and perception of EID services as well as

their experiences whiles accessing the service. All interview guides were pretested before the commencement of study.

Interviews were held in private rooms in the selected health facilities. Identification numbers rather than names were assigned to each respondent to ensure anonymity. Interviews were also audio-recorded upon approval of respondents. The interviews conducted spanned between 15 to 20 minutes.

### Data analysis

Interviews were conducted in English and two other local languages; Twi and Fante. Audio recorded interviews in English were transcribed verbatim whiles audio recordings in the local languages were translated verbatim in the local languages and re-translated into the English language for analysis.

The research team coded data with the assistance of the NVivo 11 qualitative data analysis software. The Braun & Clarke's stages of thematic analysis was used in analysing data [20]. Features of the data identified as relevant to the study objectives were coded with the assistance of the NVivo software. Codes were then collated into potential themes. All codes under each theme were reviewed to ensure coherency. Themes were reviewed continuously to ensure they accurately reflected the entire data set and appropriate names were assigned to themes.

### Ethical consideration

Ethical clearance was obtained from the Ghana Health Service Ethics Review Committee of the Research and Development Division (GHS-ERC 131/12/17). The study was introduced to the management of the selected hospitals for approval before commencement of data collection. Participation was entirely voluntary and written informed consent was obtained from respondents before interviews commenced. Respondents were informed about the purpose of the study, possible risk, benefits, confidentiality and right to refuse. The researcher also declared that she had no conflict of interest in the study.

## Results

### Socio-demographic characteristics of study participants

In all, twenty health workers who were directly involved in the delivery of EID services were interviewed, sixteen were females and the remaining males. They included Midwives, Nurses, Pharmacist, Biomedical Scientist, Data Managers, Data Officer, Models of Hope (Persons Living with HIV (PLHIV) who counsel newly diagnosed HIV mothers at the ART centres) and Psychologist. The sampled health workers age ranged from 30 to 59 years. Most had been at post for six years, with the minimum being one year and the maximum, ten years.

A total of thirty HIV positive mothers were also interviewed. The age of the mothers ranged from 20 to 49 years, with most in the 30–39 age category. Twenty-two were either married or cohabiting while the remaining had never been married. In terms of educational qualification, only two mothers interviewed had attained higher or tertiary education. Eight had acquired Secondary/ SHS/ Vocational/ Technical School education, nine had Middle School/ JSS/ JHS certificates, seven had attained primary school education, and four had no formal education.

Three of the respondents were unemployed, and twenty-one out of the remaining were traders. All the HIV positive mothers interviewed had infants who were at most a year old with the youngest aged seven weeks. *(See Table 1)*.

**Table 1. Socio-demographic characteristics of study participants.**

| Characteristics | Frequencies(N = 50) |
| --- | --- |
| **Health workers** | |
| **Sex** | |
| Male | 4 |
| Female | 16 |
| **Age (years)** | |
| 30–39 | 11 |
| 40–49 | 6 |
| 50–59 | 3 |
| **Position** | |
| Midwife | 6 |
| Nurse | 6 |
| Pharmacist | 1 |
| Biomedical Scientist | 1 |
| Data Manager | 2 |
| Data Officer | 1 |
| Model of Hope | 2 |
| Psychologist | 1 |
| **HIV Positive Mothers** | |
| **Age (years)** | |
| 20–29 | 8 |
| 30–39 | 16 |
| 40–49 | 6 |
| **Marital Status** | |
| Never married | 8 |
| Married/cohabiting | 22 |
| **Education** | |
| No education | 4 |
| Primary school | 7 |
| Middle/JSS/JHS | 9 |
| Secondary/SHS/Technical/Vocational | 8 |
| Higher/Tertiary | 2 |
| No education | 4 |
| **Occupation** | |
| Security Personnel | 2 |
| Health worker | 1 |
| Banker | 1 |
| Trader | 21 |
| Beautician | 2 |
| Unemployed | 3 |
| **Age of Exposed Infant** | |
| 6weeks– 3 months | 3 |
| 4 months– 7 months | 5 |
| 8 months– 12 months | 22 |

## Health system factors that facilitate or hinder EID service delivery

**Health workers knowledge on EID.** The study found that health workers had adequate knowledge on EID. This was evident per their responses to questions on eligibility of testing,

exact times and timelines for testing and specific tests performed on HIV exposed infants. Regarding infants eligible for testing, health workers stated that, rather than testing all newly born babies, infants of HIV positive mothers were eligible for EID.

> *"It is a little bit difficult to refer every born child to undergo testing, so mothers who are positive only are monitored and their children are also tested" (Facility A #18, Data Manager).*

> *"Every child born to an HIV positive mother is an exposed infant, so from the beginning an identified target" (Facility B #5, Nurse).*

In relation to the exact times and timelines for testing HIV exposed infants, some health workers provided responses that were in accordance with the WHO testing requirements, which recommends initial test on exposed infants within four to six weeks after birth. Some health workers indicated they had adopted a testing practice contrary to the WHO recommendation and were conducting PCR test within the first week of life.

> *"Within the first 3 to 7 days and then at 6 weeks. Initially, they were doing the EID at 6 weeks and it was changed to within the first 3 to 7 days" (Facility B #5, Nurse).*

> *"The first tests as at now is within the first week of life, the second one around 6 weeks and the third one at age one, then, at age 18 months" (Facility A #11, Nurse).*

Although health workers stated that it was a new practice that had been introduced, many health workers were unable to explain the reason for its introduction. One health worker, however, explained that the adoption of the new practice was necessitated because it was realised over the years that the number of exposed infants tested after post-natal were disproportional to the number of mothers found to be HIV positive during the antenatal period. This was because the four to six weeks testing directive by WHO was too long and it caused several exposed infants to be missed out on testing since a lot of mothers did not frequent the facility at the end of the post-natal period.

> *"With the first week of life testing, mothers will definitely be at the facility for one reason or the other, so that is an opportune time for us to get the infants" (Facility A#13, Nurse).*

Health workers were also aware that PCR (virology) and antibody (serology) tests were performed on HIV exposed infants. They were also conversant with the specific times these tests were carried out.

> *"From the 3 days to the 6 weeks we do the PCR and we do the antibody test to 1 ½ years" (Facility A # 14, Nurse).*

> *"We do the PCR at 6 weeks and the antibody test at 1 ½ years" (Facility A #13, Model of Hope).*

**Adequacy of skilled staff.** Study findings showed that health workers with sample collection skills at Facility B were inadequate, thus serving as a barrier to timely delivery of EID services.

> *". . .at some time, I think we had about just two staff trained at Antenatal Care (ANC) so if one is on leave and one is not available, mothers will come and complain that they said, they are not there so I should go and come another time" (Facility B #5, Nurse).*

On the other hand, health workers with sample collection skills at Facility A were adequate. This was primarily because the facility has devised ways of curbing the issue of staff shortages through the adoption of task shifting and task sharing strategies whereby less specialised Staff were trained to acquire skills in sample collection.

*"We train people on the job and we are doing task shifting and task sharing, so even my data manager and in some cases, cleaners and drivers effectively take sample"* (Facility A #20, Midwife).

**Availability of logistics and supplies.** Health workers in both facilities stated that DBS cards were mostly adequate and available for use. They further gave reasons for the lack of shortages, this is apparent in the quotes below.

*"We don't run out of the dry blood samples"* (Facility B #4, Midwife).

*"There is always adequate allocation by Government and also we don't wait for DBS cards to run out before we request for some"* (Facility A, #20 Midwife).

After samples were collected, properly dried and packaged, a vehicle is needed to transport samples to the reference laboratory for further processing. Health workers at Facility A articulated the frustration and difficulty they undergo to get an official vehicle to dispatch samples to the reference laboratory. Many health workers expressed the need for them to be assigned a vehicle for such purposes.

*"After taking the sample we need transport to take it to the reference lab, it has been very difficult lately, we don't get vehicle"* (Facility A #16, Data Officer).

Health workers stated that they had devised other means of transporting the samples to the reference laboratory for Polymerase Chain Reaction (PCR) testing, mainly because they did not want to impede the EID process.

*"I have a unique way of doing my work so my people have learnt it. If there is no vehicle we pick a taxi"* (Facility A #20, Midwife).

At Facility B, it was discovered that samples collected were handed over to lab technicians who frequented the Korle Bu Teaching Hospital Blood Bank twice a week, for onward delivery to the reference laboratory. A lab technician interviewed indicated that this process was tedious and recommended that a PCR machine should be made available for use in the facility.

*"There is an arrangement between the ANC and the lab so when the lab is sending some samples or whatever they have to send to Korle Bu on those designated days, they send the samples"* (Facility B #5, Nurse).

Health workers mainly ascribed EID challenges to a limited number of PCR machines available for running DNA PCR test of HIV exposed infants. They mentioned that only one PCR machine was available in the Greater Accra Region (sited at Korle Bu Teaching Hospital) and the large volumes of samples it served led to its frequent breakdown. Health workers stated that these challenges could be averted if laboratories to perform PCR analysis, as well as the number of PCR machines, are were increased.

*"The delay comes from Korle Bu, the PCR machine being only one results in the delay of results"* (Facility A #18, Data Manager).

The challenges encountered at the reference laboratory resulted in the long turnaround time (TAT) for PCR results to return to the facility and subsequently to mothers of HIV exposed infants, thus delaying the commencement of treatment for HIV infected infants. Both health workers and mothers reported that it took several months to receive PCR results. This is apparent in the quotes below.

*"Sometimes it takes up to 4 or 6 months"* (Facility B #5, Nurse).

*"After doing the test I returned after three months for the results"* (Facility B #12, Mother).

## Factors that facilitate or hinder the uptake of EID services by HIV positive mothers

**Knowledge and perception of EID.** Almost all mothers interviewed were abreast with the exact times and timelines for testing. There were, however, variations in their responses concerning the initial time for testing exposed infants. Some mothers indicated that initial testing was performed within the first week of life whiles others stated that initial testing of exposed infants was at six weeks. Mothers were also aware that an additional test was required when infants turned one and a half years.

*"After birth and 1 ½ years"* (Facility B #6, Mother).

*"When the infant turns 6 weeks, we bring the child for her to be tested for HIV another one too is done when baby is 1 ½ years"* (Facility B #1, Mother).

Mothers stated that information on EID was made available to them during PMTCT counselling sessions. Health workers affirmed these assertions; this is apparent in the quote below

*"We always counsel them (mothers) that when they deliver, even when the midwife at the maternity wards forget to tell them to go and test their babies they should also see it as a responsibility for themselves to remind the midwife"* (Facility B #5, Nurse).

Mothers perceived EID as beneficial, and they ascribed several reasons for its importance. Some mothers believed that EID helped to ascertain the status of the infants, avert disease progression and assisted in the treatment and prevention of HIV infection.

*"You need to know whether the child has it or not and that can only be through testing"* (Facility A #21, Mother).

*"To protect them so they do not get the Acquired Immunodeficiency Syndrome (AIDS)"* (Facility B #8, Mother).

*"When it is done and the result is positive we can put the child on treatment"* (Facility B #1, Mother).

*"So that you will know how to take care of the child so that you will not transfer the virus to the child"* (Facility B #7, Mother).

**Denial of HIV status.** Health workers stated that a mother's denial of her HIV status served as a significant barrier to the utilisation of EID services. They further explained that mothers who had not accepted their HIV status deemed it irrelevant to access HIV services for themselves and their exposed infants.

*"Sometimes some of the mothers are in denial, and don't want to accept that they have the condition (HIV), in such an instance, it becomes difficult to convince them to test their exposed infants since they don't see the need (Facility B #10, Psychologist).*

**EID service responsiveness.** Some health worker respondents mentioned that a lot of HIV mothers stopped frequenting the health facility to access EID services for their infants due to unavailability of health workers to take samples and delays encountered in obtaining PCR results of their exposed infants.

*"Sometimes the person to take the sample is not there and they have to come another time and then when finally, the sample are taken the results are not coming and they are frustrated so some go and they don't return" (Facility B #5, Nurse).*

Health workers further stated that these mothers were likely to return to the facility for EID services when their infants started exhibiting clinical symptoms.

*"A lot of them come back after sometime when they notice that baby is not well" (Facility B #3, Midwife).*

**Financial constraints.** Even though the provision of EID service is at no cost, other related factors such as the cost of transportation to the facility affected uptake of EID services. The greater number of mothers in the study mentioned that they did not have financial challenges while accessing EID services for their exposed infants. This was because they had either disclosed their status to their partners, family members or friends and were still obtaining financial support from such persons. Others indicated they could afford the cost of transportation from their homes to the facility since they were employed and were earning a decent income.

*"My husband is supportive but it all lies on me the mother to take care of my child so it is no cost for me at all to come here all the time" (Facility B #5, Mother).*

*"My mother is very helpful" (Facility B #1, Mother).*

Very few mothers who had not disclosed their status to anyone and were either unemployed or earning very little income could not afford the cost of transportation to the health facility. Despite these challenges, the mothers, knowing the importance of EID, did not allow their financial constraints to deter them from accessing EID for their exposed infants and, therefore sought financial assistance.

*"Often, I have financial challenges, even today I called the doctor that I don't have money and she sent me money before I came" (Facility A #17, Mother).*

**The attitude of health workers towards HIV positive mothers.** All mothers interviewed stated that health workers relationship with them were cordial and applauded them for their positive attitudes. Some mothers, however, indicated that health workers only exhibited

unfriendly attitudes or scolded them when they reported late to the health facility or retuned after defaulting for several months.

*"Nurses are very good, the way they relate with us like we are just like them" (Facility A # 20, Mother).*

Mothers further commended health workers for improving their knowledge of HIV and EID related issues during PMTCT counselling sessions. Mothers stated that the support, encouragement and information provided to them by health workers during the antenatal period made them appreciate the importance of HIV testing services. This further motivated them to continue accessing HIV services for themselves as well as utilise EID services for their exposed infants after birth.

*"they (health workers) have been so helpful, when they tested me and I was positive in fact I thought that was the end of my life, but they told me once I take my drugs well I will be fine like every normal person and baby will be fine too. . .they constantly reminded us during counselling sessions to bring baby for prophylaxis and testing after birth" (Facility A #16, Mother).*

## Discussion

### Health system factors that facilitate or hinder EID service delivery

**Health workers knowledge on EID.** Findings from this study agrees with results of Ghanaian research which showed that health workers had adequate knowledge on EID of HIV services and had insights on the eligibility of testing, exact times and timelines for testing HIV exposed infants [21]. Study finding is however contrary to studies carried out in South Africa and other sub-Saharan Africa countries which reported health worker's inadequate knowledge on EID due to their inability to indicate the number, types of test and exact times and timelines for performing tests for HIV exposed infants [21,22].

Importantly, this study found that health workers practised after-birth testing which contradicts the 4 to 6 weeks testing directive by the WHO. This finding also disagrees with Ghana's national testing guidelines which recommends PCR test for exposed infants at six weeks. Although this study attributed the reason for after-birth testing to the avoidance of mother-infant pair being missed for testing after the post-natal period, a recent study conducted in Lesotho revealed the likelihood of infants infected in utero to have suppressed immunity or die from HIV related factors by six weeks [12]. Studies carried out in South Africa also found that HIV infection had significantly progressed in 62% of infants who tested at six weeks before commencement of ART at a median age of 8.4 weeks [12]. Prompt initiation of HIV infected infants on ART increases their chances of survival [23], therefore delaying infant testing until six weeks may result in late ART initiation. Another study also found that HIV infected infants who tested at birth, obtained their PCR results earlier and were able to start ART earlier as compared to those who tested at six weeks [24].

**Adequacy of skilled staff.** The results of the study are in line with findings of previously conducted studies which reported that the number of trained health workers with requisite skills in DBS sample collection was limited thus affecting EID service provision [8,25]. Some health workers in this study, however, indicated that they had devised a method of dealing with the issue of staff shortages through the adoption of task shifting practices whereby less specialised Staff in the health facility were trained in sample collection. Similar studies

conducted in Malawi also suggest task shifting as a means of addressing the issue of limited trained Staff and workload [26]. It is vital that staff involved in task shifting are adequately trained and under supervision to ensure quality healthcare provision [27].

**Availability of logistics and supplies.** Earlier studies in Kenya, Zambia and Burkina Faso reported stock out of DBS cards as a constraining factor in delivering EID services [8,13,28]. Results of this study are however at variance with these earlier findings as DBS cards were reportedly available due to adequate allocation to the study facilities by Government as well as proactivity on the part of health workers to ensure all-time availability.

Findings of this study are consistent with other studies which reported challenges in obtaining vehicles to transport samples to the reference laboratory [8,16] due to inadequate and frequent breakdown of vehicles [29]. Providing these EID centres with adequate vehicles and motorcycles will facilitate the transportation of samples to the reference laboratory for processing [30].

Study results like previous studies conducted in sub-Saharan Africa disclosed that limited laboratories to perform PCR testing, limited PCR machines and frequent breakdown of these machines contributed to the delay in return of results of HIV exposed infants to the health facility and eventually to mothers [7,13]. This study identified that only one PCR machine is available in the study region, and all samples collected from various health facilities that perform EID were sent to the reference laboratory for processing. These large volumes of sample analysis resulted in the frequent break down of the machine resulting in delays in the EID process. Government should allocate funds for the provision of additional PCR machines and ensure that these machines are strategically positioned in selected health facilities.

Although the WHO recommends that PCR results of HIV exposed infants should be returned to the facility and ultimately to mothers at most within four weeks after samples have been collected [31], this study reported that it took several months for PCR results to return to the health facility and subsequently the mother. This finding is, however, similar to a Zambian study which reported 92 days as turnaround time for receipt of results [12]. Other studies have also reported the long turnaround time for receipt of PCR results as an impediment in the EID process and resultant delay in ART initiation among infants found to be HIV positive [32,33]. Initiating Point-of-Care (POC) testing services will allow rapid receipt of test results and enable prompt initiation of ART for infected infants [34–36].

## Factors that facilitate or hinder the uptake of EID services by HIV positive mothers

**Knowledge and perception of EID.** Contrary to the result of some recent studies [30,37], this study showed that HIV positive mothers were satisfactorily knowledgeable on issues related to EID. They provided suitable responses to questions on specific tests conducted on their infants and the exact times these tests were performed. A Kenyan study attributed inadequacy in knowledge of mothers to the failure of health workers to provide mothers with sufficient information on EID during ANC and PMTCT training [13]. In this study however, mothers interviewed stated that details on EID were made available to them during PMTCT training sessions, suggesting the sterling efforts put in place by health workers to provide adequate health information since maternal awareness of EID services increases the chances of uptake of such EID services [38].

Findings from this study are comparable to the results of a Malawian study which reported that mothers were aware of the availability of EID services [39]. It corroborates with the results of a South African study, where mothers perceived EID as beneficial [40]. Results of studies conducted in Zambia and Mozambique respectively, revealed that mothers who were adhering

to PMTCT and on ART had prior knowledge of HIV hence were more likely to utilise EID services [16,41]

**Denial of HIV status.** Consistent with findings from other studies, factors such as a mother's non-acceptance or denial of her HIV status served as a barrier to the uptake of EID services [38,42]. Mothers in denial must be counselled persuasively on the need to accept their current status and the importance of testing their exposed infants. In addition, other HIV infected mothers who have come to terms with their diagnosis should be encouraged to share their experiences with newly diagnosed mothers living in denial [6]. Without early infant diagnosis, however, infected infants cannot be identified and commencement of ART becomes impossible, thus decreasing chances of infant survival [1,2].

**EID service responsiveness.** Findings agree with a previous study which ascribed unsuccessful visits to the health facility by mothers to access EID services for their exposed infants and low service responsiveness as a reason for mother's reluctance to accessing EID services [43]. The long turnaround time for receipt of PCR results has also been identified as a significant factor for mother's non-completion of the EID process. This is because mothers may be discouraged from returning to the facility to collect results of their infants after several fruitless visits [14,44].

In instances were test results of exposed infants are not ready on the date provided, health workers should initiate contact with mothers via phone calls and SMS to reschedule hospital appointments [14] and to promptly notify mothers to visit the facility when results are available. Additionally, a toll-free number should be provided to all mothers to call periodically to check the status of their infant's results.

Similar to findings of a Kenyan study, this present study found that mothers were likely to return to the facility for EID services once infants exhibited clinical symptoms [13].

**Financial constraints.** Unlike previously conducted studies which attributed the cost of transportation as a barrier to the uptake of EID by mothers [8], transport cost was not a hindrance in this current study.

This is because most mothers in the study had financial independence since they were working and earning income or were obtaining support from people they had disclosed their status to. This study also found that a few mothers who had financial difficulties because they were unemployed or had not disclosed their status to anyone still found a way to access EID services for their infants since they were aware of the benefits of EID. Study finding agree to a research carried out in Mozambique which found that mothers who were employed and earning decent salaries did not go through any financial hardship in accessing EID services for their exposed infants [41].

**The attitude of health workers towards HIV positive mothers.** This study findings, like several others, emphasises the importance of health workers to exhibit positive attitudes to patients during ANC and PMTCT sessions. This is because the attitude of health workers towards mothers contributed to mothers decision to return for EID services after delivery or otherwise [2]. A Kenyan study also reported that brewing cordial relationships between health workers and clients increased maternal uptake of services for their HIV exposed infants [13]. Study finding disputes results of a previously conducted research which reported abuse and discrimination of mothers by health workers during ANC services because of their HIV status [45,46].

## Study limitations

1. Since the interviews were conducted during work hours, researchers structured study tools for a minimum of 15–20 minutes so not to interrupt work schedules. This, however, in no way affected the quality of responses from respondents.

2. The study covered only HIV positive mothers enrolled in PMTCT. This could have introduced some biases, therefore a limitation to the study.

3. Just as in the case of HIV positive mothers, the health workers sampled were directly involved in the delivery of EID services, thus findings cannot be generalised to the population of general health work force in Ghana.

## Conclusion

In conclusion, study findings attributed low coverage of EID services to both maternal and health system factors. Health system factors, notably, the limited number and frequent breakdown of the PCR machine resulted in the long turnaround time of receipt of results, making early initiation of HIV infected infants on ART impossible.

Other constraints identified were inadequate Staff with sample collection skills and unavailability of vehicles to convey samples to the reference laboratory. Maternal factors such as the denial of HIV status and lost to follow up due to frustrations encountered in accessing service and delay in the return of PCR results served as barriers to mother's utilisation of EID services for their exposed infants.

The study also found that adequate knowledge of Staff on EID, availability of DBS cards and the adoption of task shifting strategies facilitated EID service delivery. Knowledge of EID, perceived importance of EID, financial support and independence as well as positive attitudes of health workers significantly contributed to mother's uptake of EID services.

The factors attributing to the low coverage of EID must be urgently addressed to improve service delivery and uptake.

## Supporting information

**S1 File. Health workers transcript.**
(PDF)

**S2 File. HIV mothers transcript.**
(PDF)

**S3 File. ISSM_COREQ_Checklist.**
(PDF)

## Acknowledgments

The authors would like to thank the management of the study sites for granting us permission to use their facilities for data collection. The authors would also like to thank the health workers and HIV positive mothers who willingly availed themselves to participate in the study.

## Author Contributions

**Conceptualization:** Antoinette Kailey Ankrah, Phyllis Dako-Gyeke.

**Data curation:** Antoinette Kailey Ankrah.

**Methodology:** Antoinette Kailey Ankrah, Phyllis Dako-Gyeke.

**Resources:** Antoinette Kailey Ankrah.

**Supervision:** Phyllis Dako-Gyeke.

**Writing – original draft:** Antoinette Kailey Ankrah.

**Writing – review & editing:** Phyllis Dako-Gyeke.

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
