## [Decision Letter · Decision Letter 0]

30 Jun 2020

PONE-D-20-16385

Factors influencing use of early infant diagnosis of HIV services in Greater Accra, Ghana: a qualitative study

PLOS ONE

Dear Dr. Ankrah,

Thank you for submitting your manuscript to PLOS ONE. After careful consideration, we feel that it has merit but does not fully meet PLOS ONE’s publication criteria as it currently stands. Therefore, we invite you to submit a revised version of the manuscript that addresses the points raised during the review process.

We look forward to receiving your revised manuscript.

Kind regards,

Emma Sacks

Academic Editor

PLOS ONE

Additional Editor Comments:

This is an important topic, but a number of edits are required to strengthen the analysis methods and presentation of results. Please see the comments below and from the reviewer. Thank you.

Overall, a nicely done study which addresses an important issue of EID, but many revisions are required.

• The section on “consequences after disclosure” needs to be rewritten to answer the primary question about barriers to EID services. Does the disclosure dissuade people from testing? If so, this needs to be supported with data. Similarly, does the “attitude of health workers” encourage earlier care seeking, or does it discourage seeking later?

• Given that there are only 2 hospitals and the authors are comparing practices, consider using pseudonyms. Further, are there funding/resource differences between the hospitals? It would be good to understand better why one hospital was able to innovate more than the other.

• Is it better to discuss risk reduction rather than prevention of transmission where possible

• Is the timing of the 6 week visit aligned with a vaccine visit?

• Line 68: please define “HIV advancement”

• Not only are only 54% of exposed infants tested, but an even smaller percent receive results

• Line 96: what is the age range of this prevalence?

• What is “Models of hope”? (text and tables)

• Can you please format the table to be smaller (should fit on less than one page)?

• If the participants were only those who were enrolled in PMTCT, there may be bias because those who disengaged from care or did not seek care are not included. Please be sure to include this is the limitation.

• If the sample size was determined by evaluating data saturation, then was data analysis conducted simultaneous to collection, or was another process like interim debriefings used?

• 15-20 min is very short for an interview. Can the authors explain why so short, and perhaps include this in the limitations.

• Line 130 please list the languages

• Lines 224-226 should be moved to the discussion

• Line 244: was the “policy” on testing at birth something decided by each facility, or something happening commonly across the region/country?

• WHO has recommended that official birth testing programs only begin once a 6 week program has been well established; are there concerns about the impact of birth testing on the 6 week program

• Some studies on birth testing should be cited: for example:

o https://pubmed.ncbi.nlm.nih.gov/27540110

o https://pubmed.ncbi.nlm.nih.gov/32183751

o https://pubmed.ncbi.nlm.nih.gov/32329186

o https://pubmed.ncbi.nlm.nih.gov/32520911

• Line 284: instead of “the study” please say who pointed out the barrier

• Line 308: which clinical symptoms are cognized by families? And at what age of infants?

• Line 345: this is unclear if there is a cost or not. Similary, in line 474, was the only cost about transport or are there other costs?

• Line 351-353: this could be clearer: is the barrier only about the importance of testing, regardless of finance? This seems to contradict the previous.

• Line 390: the information about cleaners and driver is new information, so should first be in the results before the discussion.

• Line 391: what similar findings? From where?

• Line 413: the use of the word “average” is too quantifiable for this paper. Better to say something like “many people reported”

• Line 452: presenting with clinical symptoms may be too late; it would be helpful to understand more about the severity of symptoms and age of presentation

• Line 457: how do women manage testing and treatment when they have not disclosed?

• Like 460: what about other family members, for example grandparents?

• Line 482: may want to cite additional literature about “Respectful care”

• It’s a nice finding that the staff have been able to be creative; perhaps this could be further emphasized in the discussion.

• Please be careful when citing a rate – not every prevalence is a rate

• Citation #32 should be replaced with a peer review article, for example: https://linkinghub.elsevier.com/retrieve/pii/S2352-3018(19)30033-5

• While this is not a quantitative study, it is helpful to include a bit more about when the respondents fully agreed vs where there was more disagreement

• The paper requires some grammar editing; there are many sentences which should be more clearly written for a scientific publication

3. We note you have included a table to which you do not refer in the text of your manuscript. Please ensure that you refer to Table 1 in your text; if accepted, production will need this reference to link the reader to the Table.

Reviewers' comments:

Reviewer's Responses to Questions

**Comments to the Author**

1. Is the manuscript technically sound, and do the data support the conclusions?

Reviewer #1: No

2. Has the statistical analysis been performed appropriately and rigorously? 

Reviewer #1: N/A

3. Have the authors made all data underlying the findings in their manuscript fully available?

Reviewer #1: No

4. Is the manuscript presented in an intelligible fashion and written in standard English?

Reviewer #1: Yes

5. Review Comments to the Author

Reviewer #1: The authors explore an important and current issue in on-going battle against HIV: ensuring the service capacity and patient uptake of early infant diagnosis. Overall, the article needs great improvement to clarify the methods of analysis, central objectives, and how the categories of results presented connect to an overarching thesis. It will be a lot of work, but this is an important topic, so I hope the authors are up for the challenge. The major revisions I suggest are:

Intro

1. The data on lowering the risk of MTCT with pre-natal ART is not clearly connected to the central topic of the paper: ensuring EID after birth. The statistics on MTCT in the abstract and the intro get confusing, and I would recommend establishing a stronger focus on EID right at the beginning.

2. The intro could be strengthened with some review of the literature from other sub-Saharan African countries. Much of this is reviewed in the discussion. I would recommend moving some of the literature to the intro that helps establish the goal of the paper and lays out what is already known on health care and maternal factors and why the context of Ghana might be different and in need of further study.

3. The final sentence in the intro states the goal of the study. I am wondering about the word “use” as it does not seem as if the authors are looking at EID use only, but also functionality of services. Indeed, it is unclear how many of the themes in the results relate to use (e.g. Do you have evidence that the availability of logistics and supplies is affecting use?) I would carefully reword this objective sentence and ensure that all results clearly relate to the stated objective.

4. There is also an important distinction between initiating EID and completing the full EID cascade. Please describe the EID cascade according to Ghanaian guidelines and clarify which outcome (initiation or completion) relates to the literature you review and the objective of your study.

Methods:

5. There are many essential details of the methods missing. I strongly recommend the authors reference the COREQ checklist in their reporting and include the checklist in their resubmission. http://cdn.elsevier.com/promis_misc/ISSM_COREQ_Checklist.pdf. Importantly, the authors need to be more about what qualitative approach was used (Was there a theoretical framework? Did the authors intend use grounded theory or to build a theory?)

6. The authors state they used purposive sampling, but not the criteria that guided the sampling. How was it purposive? Was there a specific type of purposive sampling you employed (e.g. maximum diversity, criterion, etc.) What sort of representation were you aiming to achieve? Was there other formal eligibility criteria (e.g. age).

Results

7. I recommend reworking the results section once the objective and theoretical framework are clarified. Each sub-section of the results needs to clearly feed into the central objective. Right now, it if unclear how certain sections work into the goal of assessing facilitators and barriers of EID use. For example, “Lost to follow-up” seems more like a consequence, not a barrier/facilitator; “Disclosure of HIV status” is not clearly connected to EID in the data presented

8. I’m unsure if the quote in “Denial of HIV Status” is saying that this is a barrier or a facilitator of EID.

9. I would recommend if the “Adequacy of skilled staff” section remains to clarify that you are assessing perceptions of adequacy of skilled staff, not actually using any formal criteria.

10. How does the turnaround time relate to the steps in the EID cascade? The months of waiting referenced- was for a final result (and completion of the cascade) or for the result of the more recent test? There is a big difference here.

Discussion:

11. Please include a Limitations section

12. I am curious why the authors chose the subheading for the Discussion as they do not correspond to the stated goals of the study nor the organization of the results section.

13. There are often some results stated in the Discussion that were not stated in the Results section. Please ensure the Discussion interprets results already reported and does not bring in new findings.

14. Consider reviewing the following recent literature that gives more depth into:

o System-level interventions for EID

Finocchario-Kessler, S., Gautney, B., Cheng, A., Wexler, C., Maloba, M., Nazir, N., Khamadi, S., Lwembe, R., Brown, M., Odeny, T.A., Dariotis, J.K., Sandbulte, M., Mabachi, N., Goggin, K., 2018. Evaluation of the HIV Infant Tracking System (HITSystem) to optimise quality and efficiency of early infant diagnosis: a cluster-randomised trial in Kenya. Lancet HIV. https://doi.org/10.1016/S2352-3018(18)30245-5

Schmitz, K., Basera, T.J., Egbujie, B., Mistri, P., Naidoo, N., Mapanga, W., Goudge, J., Mbule, M., Burtt, F., Scheepers, E., Igumbor, J., 2019. Impact of lay health worker programmes on the health outcomes of mother-child pairs of HIV exposed children in Africa: A scoping review. PLoS One. https://doi.org/10.1371/journal.pone.0211439

o Complexities of the EID Casade

Mofenson, L. M., Cohn, J., & Sacks, E. (2020). Challenges in the Early Infant HIV Diagnosis and Treatment Cascade. JAIDS Journal of Acquired Immune Deficiency Syndromes, 84, S1-S4.

o Motherhood identify and EID completion

Hurley, E. A., Odeny, B., Wexler, C., Brown, M., MacKenzie, A., Goggin, K., ... & Finocchario-Kessler, S. (2020). “It was my obligation as mother”: 18-Month completion of Early Infant Diagnosis as identity control for mothers living with HIV in Kenya. Social Science & Medicine, 112866.

6. PLOS authors have the option to publish the peer review history of their article (what does this mean?). If published, this will include your full peer review and any attached files.

Reviewer #1: No

---

## [Author Response · Author response to Decision Letter 0]

7 Sep 2020

Editors Comments 

1. The section on “consequences after disclosure” needs to be rewritten to answer the primary question about barriers to EID services. Does the disclosure dissuade people from testing? If so, this needs to be supported with data. Similarly, does the “attitude of health workers” encourage earlier care seeking, or does it discourage seeking later?

Response -The section of consequence after disclosure has been deleted since it does not direct linkage with the objectives of the study

2. Given that there are only 2 hospitals and the authors are comparing practices, consider using pseudonyms. Further, are there funding/resource differences between the hospitals? It would be good to understand better why one hospital was able to innovate more than the other.

Response -Pseudonyms have been assigned to the two hospitals as Facility A and B. Facility A and B have described and compared as Regional and District Hospitals

3. Is it better to discuss risk reduction rather than prevention of transmission where possible

Response -This comment is well noted.

4. Is the timing of the 6 week visit aligned with a vaccine visit? 

Response -Yes, the timing of 6 weeks is aligned to a vaccine visit, however, this study did not assess any linkage between EID service and uptake of immunisation. 

5. Line 68: please define “HIV advancement” 

Response -This has been changed for clarity.

6. Not only are only 54% of exposed infants tested, but an even smaller percent receive results 

Response -This comment has been incorporated into the paper

7. Line 96: what is the age range of this prevalence? 

Response -The age range of this prevalence is 15- 49yrs

8. What is “Models of hope”? (text and tables) 

Response - Models of Hope are Persons Living with HIV who counsel newly diagnosed HIV mothers at the ART centres

9. Can you please format the table to be smaller (should fit on less than one page)? 

Response -The table has been formatted accordingly

10. If the participants were only those who were enrolled in PMTCT, there may be bias because those who are disengaged from care or did not seek care are not included. Please be sure to include this in the limitation. 

Response -We agree with this comment and have included it in the limitation.

11. If the sample size was determined by evaluating data saturation, then was data analysis conducted simultaneous to collection, or was another process like interim debriefings used? 

Response -Saturation point was reached with periodic debriefing among the research team

12. 15-20 min is very short for an interview. Can the authors explain why so short, and perhaps include this in the limitations. Since the interviews were conducted during working hours, it was structured for a minimum of 15-20 mins in order for minimum interruption of work and time. 

Response -This has been included to the limitations.

13. Line 130 please list the languages 

Response -Languages have been included as Twi and Fante 

14. Lines 224-226 should be moved to the discussion

Response -This has been moved to the discussion section

15. Line 244: was the “policy” on testing at birth something decided by each facility, or something happening commonly across the region/country? 

Response - Though birth testing is not a national policy, it is a common practice in both facilities utilised for the study. 

16. WHO has recommended that official birth testing programs only begin once a 6 week program has been well established; are there concerns about the impact of birth testing on the 6 week program

Response -This study did not assess this component.

17.Some studies on birth testing should be cited: for example:

o https://pubmed.ncbi.nlm.nih.gov/27540110

o https://pubmed.ncbi.nlm.nih.gov/32183751

o https://pubmed.ncbi.nlm.nih.gov/32329186

o https://pubmed.ncbi.nlm.nih.gov/32520911

Response -Some of the suggested studies on birth testing have been included to the study

18. Line 284: instead of “the study” please say who pointed out the barrier 

Response -Changes have been effected accordingly

19. Line 308: which clinical symptoms are recognized by families? And at what age of infants?

Response -Though this was mentioned by the health workers, the details of the clinical symptoms was not enquired.

20. Line 345: this is unclear if there is a cost or not. Similarly, in line 474, was the only cost about transport or are there other costs? 

Response -Though the provision of EID services is at no cost, other indirect cost such as transportation to the Facility was discussed.

21. Line 351-353: this could be clearer: is the barrier only about the importance of testing, regardless of finance? This seems to contradict the previous.

Response -The wording has been improved to remove the seeming contradiction.

22. Line 390: the information about cleaners and driver is new information, so should first be in the results before the discussion. 

Response - Information about cleaners and drivers have been included in the result section as proposed.

23. Line 391: what similar findings? From where? 

Response -We have mentioned the country from where a similar study on task shifting was reported 

24. Line 413: the use of the word “average” is too quantifiable for this paper. Better to say something like “many people reported” 

Response - Changes have been effected accordingly

25. Line 452: presenting with clinical symptoms may be too late; it would be helpful to understand more about the severity of symptoms and age of presentation 

Response -This comment is well noted.

26. Line 457: how do women manage testing and treatment when they have not disclosed? 

Response -The study did not assess this component. 

27. Like 460: what about other family members, for example grandparents? 

Response -We note this comment, however, grandparents were not mentioned by the respondents

28. Line 482: may want to cite additional literature about “Respectful care”

Response -Literature about Respectful care cited

29. It’s a nice finding that the staff have been able to be creative; perhaps this could be further emphasized in the discussion. 

Response -The creativity of the Staff have been further discussed in the relevant sections.

30. Please be careful when citing a rate – not every prevalence is a rate 

Response -This comment is well noted.

31. Citation #32 should be replaced with a peer review article, for example: https://linkinghub.elsevier.com/retrieve/pii/S2352-3018(19)30033-5

Response- Citation has been revised.

32. While this is not a quantitative study, it is helpful to include a bit more about when the respondents fully agreed vs where there was more disagreement 

Response -This comment is well noted.

33. The paper requires some grammar editing; there are many sentences which should be more clearly written for a scientific publication 

Response -This paper has been thoroughly reviewed

COMMENTS FROM REVIEWER 1 

1.The data on lowering the risk of MTCT with pre-natal ART is not clearly connected to the central topic of the paper: ensuring EID after birth. The statistics on MTCT in the abstract and the intro get confusing, and I would recommend establishing a stronger focus on EID right at the beginning. 

Response - This comment is well noted. EID has been introduced right from the onset of the paper.

2. The intro could be strengthened with some review of the literature from other sub-Saharan African countries. Much of this is reviewed in the discussion. I would recommend moving some of the literature to the intro that helps establish the goal of the paper and lays out what is already known on health care and maternal factors and why the context of Ghana might be different and in need of further study.

Response - This comment is well noted, changes have been effected accordingly

3. The final sentence in the intro states the goal of the study. I am wondering about the word “use” as it does not seem as if the authors are looking at EID use only, but also functionality of services. Indeed, it is unclear how many of the themes in the results relate to use (e.g. Do you have evidence that the availability of logistics and supplies is affecting use?) I would carefully reword this objective sentence and ensure that all results clearly relate to the stated objective. 

Response -This comment is well noted, we have modified the objective of the study to focus on the factors that facilitate or hinder the delivery and uptake of EID of HIV services. Based on this revision, we have consequently reworded the title of the study.

4. There is also an important distinction between initiating EID and completing the full EID cascade. Please describe the EID cascade according to Ghanaian guidelines and clarify which outcome (initiation or completion) relates to the literature you review and the objective of your study. 

Response -This comment is noted. The EID cascade based on Ghanaian guidelines, which we believe conforms to others includes;

1. Identification of exposed infant

2. EID Test

3. Return of PCR Results to Health Facility and Mother 

4. If Infant is Positive – Link to Care /Initiation of ART

5. If Infant is Negative (not breastfeeding) – Reassure Family

6. If Infant is Negative (breastfeeding) – Plan rapid weaning at 6 -12 months

7. Carry out Anti-body test when infant is one and half years.

The obvious outcome is therefore completion of the cascade.

5. There are many essential details of the methods missing. I strongly recommend the authors reference the COREQ checklist in their reporting and include the checklist in their resubmission. http://cdn.elsevier.com/promis_misc/ISSM_COREQ_Checklist.pdf. Importantly, the authors need to be more about what qualitative approach was used (Was there a theoretical framework? Did the authors intend use grounded theory or to build a theory?) 

Response -The COREQ Checklist has been used in reporting and a copy of the checklist has been the resubmission

6. The authors state they used purposive sampling, but not the criteria that guided the sampling. How was it purposive? Was there a specific type of purposive sampling you employed (e.g. maximum diversity, criterion, etc.) What sort of representation were you aiming to achieve? Was there other formal eligibility criteria (e.g. age). 

Response -Purposive sampling and Criterion sampling were utilised in the studies. This has been elaborated in the methods section

7. I recommend reworking the results section once the objective and theoretical framework are clarified. Each sub-section of the results needs to clearly feed into the central objective. Right now, it if unclear how certain sections work into the goal of assessing facilitators and barriers of EID use. For example, “Lost to follow-up” seems more like a consequence, not a barrier/facilitator; “Disclosure of HIV status” is not clearly connected to EID in the data presented

Response - This comment is noted. Amendments have been made to the result section to reflect the central objectives of the study. 

8. I’m unsure if the quote in “Denial of HIV Status” is saying that this is a barrier or a facilitator of EID. 

Response -The write up on Denial of HIV status has been revised for clarity. The quote however was maintained

9. I would recommend if the “Adequacy of skilled staff” section remains to clarify that you are assessing perceptions of adequacy of skilled staff, not actually using any formal criteria. 

Response -This comment is well noted

10. How does the turnaround time relate to the steps in the EID cascade? The months of waiting referenced- was for a final result (and completion of the cascade) or for the result of the more recent test? There is a big difference here.

Response -Several outcomes are anticipated in a long waiting time for mothers. These mothers may either drop out and not complete the EID process or continue to wait for the return of the results.

11. Please include a Limitations section 

Response -A limitation section has been included 

12. I am curious why the authors chose the subheading for the Discussion as they do not correspond to the stated goals of the study nor the organization of the results section.

Response -The subheadings of the Discussion section have been revised to correspond with that of the Results section 

13. There are often some results stated in the Discussion that were not stated in the Results section. Please ensure the Discussion interprets results already reported and does not bring in new findings.

Response -All information reported in the Results section have now been incorporated in the Discussion section

14.Consider reviewing the following recent literature that gives more depth into:

o System-level interventions for EID

Finocchario-Kessler, S., Gautney, B., Cheng, A., Wexler, C., Maloba, M., Nazir, N., Khamadi, S., Lwembe, R., Brown, M., Odeny, T.A., Dariotis, J.K., Sandbulte, M., Mabachi, N., Goggin, K., 2018. Evaluation of the HIV Infant Tracking System (HITSystem) to optimise quality and efficiency of early infant diagnosis: a cluster-randomised trial in Kenya. Lancet HIV. https://doi.org/10.1016/S2352-3018(18)30245-5

Schmitz, K., Basera, T.J., Egbujie, B., Mistri, P., Naidoo, N., Mapanga, W., Goudge, J., Mbule, M., Burtt, F., Scheepers, E., Igumbor, J., 2019. Impact of lay health worker programmes on the health outcomes of mother-child pairs of HIV exposed children in Africa: A scoping review. PLoS One. https://doi.org/10.1371/journal.pone.0211439

o Complexities of the EID Casade

Mofenson, L. M., Cohn, J., & Sacks, E. (2020). Challenges in the Early Infant HIV Diagnosis and Treatment Cascade. JAIDS Journal of Acquired Immune Deficiency Syndromes, 84, S1-S4.

o Motherhood identify and EID completion

Hurley, E. A., Odeny, B., Wexler, C., Brown, M., MacKenzie, A., Goggin, K., ... & Finocchario-Kessler, S. (2020). “It was my obligation as mother”: 18-Month completion of Early Infant Diagnosis as identity control for mothers living with HIV in Kenya. Social Science & Medicine, 112866.

Response - Comment well noted.

---

## [Decision Letter · Decision Letter 1]

26 Oct 2020

PONE-D-20-16385R1

Factors influencing the delivery and uptake of early infant diagnosis of HIV services in Greater Accra, Ghana: a qualitative study

PLOS ONE

Dear Dr. Ankrah,

Thank you for submitting your manuscript to PLOS ONE. After careful consideration, we feel that it has merit but does not fully meet PLOS ONE’s publication criteria as it currently stands. Therefore, we invite you to submit a revised version of the manuscript that addresses the points raised during the review process.

We look forward to receiving your revised manuscript.

Kind regards,

Emma Sacks

Academic Editor

PLOS ONE

Additional Editor Comments (if provided):

The authors have done a commendable job in addressing the reviewer and editor comments. The objectives and overall focus are much clearer. However, there are some additional minor points to address. Please see the further comments from the reviewer. In particular, it is unclear if the authors used the COREQ checklist, and we feel it would be helpful for ensuring that the various aspects of qualitative methodology are covered.

Reviewers' comments:

Reviewer's Responses to Questions

**Comments to the Author**

1. If the authors have adequately addressed your comments raised in a previous round of review and you feel that this manuscript is now acceptable for publication, you may indicate that here to bypass the “Comments to the Author” section, enter your conflict of interest statement in the “Confidential to Editor” section, and submit your "Accept" recommendation.

Reviewer #1: (No Response)

2. Is the manuscript technically sound, and do the data support the conclusions?

Reviewer #1: Yes

3. Has the statistical analysis been performed appropriately and rigorously? 

Reviewer #1: N/A

4. Have the authors made all data underlying the findings in their manuscript fully available?

Reviewer #1: No

5. Is the manuscript presented in an intelligible fashion and written in standard English?

Reviewer #1: Yes

6. Review Comments to the Author

Reviewer #1: I see that the authors have made many revisions to their manuscript. I especially appreciate the clarification of the objective and the reorganization of the intro, results, and headings of the discussion. I have some new comments on the draft and am requesting some clarifications on the responses to my original comments:

New comment: It would be wonderful if the new objective was reflected in the abstract.

New comment: Second paragraph of the discussion, it is not clear to me if you are suggesting the WHO revise it’s guidelines? I would word this carefully, as your data does not directly suggest guidelines should be revised, but that people are practicing earlier testing in other settings and calling into question the guidelines.

New comment: Please reflect on the generalizability of the findings. You allude to limitations of generalizability in the mothers you sampled. Was there anything about the health care workers or sites you collected data from that allows or doesn’t allow you to generalize results to other settings/populations? Expand on any limits to generalizability in the limitations section.

New comment: The discussion is much improved, but still lacks some key take-aways in terms of public health recommendations. Some of the literature I suggested in my previous comment #14 on system-level interventions might be worth reflecting on, or any other comments the authors want to make about what their findings mean for developing better EID systems.

Original comment #2: The intro has been much improved, especially since the objective has now been widened to include “facilitators and barriers”. However, the authors need to point out the gaps in the literature that their study sought to address. If all of this is known about facilitators and barriers to EID, why conduct the study? It may be that simply no study like this has been conducted in the Ghanaian context, in which case, the authors should point out.

Original comment #4: When I asked for clarification of the outcome, I meant the health behavior outcome of interest in this study. I see you said “the obvious outcome is therefore the completion of the cascade” but it does seem like most of the research inquiry is focused on general uptake of EID (beginning the process). I think that would be important to clarify that you were interested in facilitators and barriers both to initiating EID and to completing it once started. Adding details of the EID cascade into the text would be helpful too- the end of the first paragraph of the introduction is currently misleading in making it seem like the virologic test within 4-6 weeks is all that encompasses EID.

Original comment #5: I don’t see the checklist but perhaps the editor does. In any case, it does look like the authors incorporated much of these criteria. Thank you.

Original comment #6: I appreciate the authors saying they used “criterion sampling” but it would be beneficial to understand what the criterion was they were sampling on (mothers with certain characteristics, experiences…?) Saying you used criterion sampling is good, but you need to go a step further and elaborate on the criteria applied. You also mention “eligibility criteria” so should define that somewhere as well.

Original comment #7: The section looks better, but the quote is still a little difficult to comprehend (“so they seeking treatment…”). Can the syntax be improved? (perhaps “so [the idea of] seeking treatment…”)

7. PLOS authors have the option to publish the peer review history of their article (what does this mean?). If published, this will include your full peer review and any attached files.

Reviewer #1: No

---

## [Author Response · Author response to Decision Letter 1]

7 Dec 2020

Comment 1- It would be wonderful if the new objective was reflected in the abstract.

Response - This comment is well noted. The abstract has been revised to include the objectives of the study.

Comment 2 - Second paragraph of the discussion, it is not clear to me if you are suggesting the WHO revise it’s guidelines? I would word this carefully, as your data does not directly suggest guidelines should be revised, but that people are practicing earlier testing in other settings and calling into question the guidelines. 

Response - This comment is well noted. The statement on WHO revising its guidelines has been deleted to avoid ambiguity.

Comment 3 -Please reflect on the generalizability of the findings. You allude to limitations of generalizability in the mothers you sampled. Was there anything about the health care workers or sites you collected data from that allows or doesn’t allow you to generalize results to other settings/populations? Expand on any limits to generalizability in the limitations section.

Response - Like HIV positive mothers, health workers were purposively selected for the study. These health workers are directly involved in the delivery of EID services thus it will not be possible to generalise findings to the population general health work force in Ghana. This has been included to the study limitations.

Comment 4 - The discussion is much improved, but still lacks some key take-aways in terms of public health recommendations. Some of the literature I suggested in my previous comment #14 on system-level interventions might be worth reflecting on, or any other comments the authors want to make about what their findings mean for developing better EID systems.

Response -This comment is well noted and revisions have been made accordingly

These revisions can be seen on; Line 439 – 442, Line 452 – 454, Line 462 – 463, Line 471 – 472,Line 496 – 500, Line 511 - 515

Comment 5 - The intro has been much improved, especially since the objective has now been widened to include “facilitators and barriers”. However, the authors need to point out the gaps in the literature that their study sought to address. If all of this is known about facilitators and barriers to EID, why conduct the study? It may be that simply no study like this has been conducted in the Ghanaian context, in which case, the authors should point out. 

Response - This comment is well noted. Changes have been effected in the introduction accordingly.

Comment 6 -When I asked for clarification of the outcome, I meant the health behavior outcome of interest in this study. I see you said “the obvious outcome is therefore the completion of the cascade” but it does seem like most of the research inquiry is focused on general uptake of EID (beginning the process). I think that would be important to clarify that you were interested in facilitators and barriers both to initiating EID and to completing it once started. Adding details of the EID cascade into the text would be helpful too- the end of the first paragraph of the introduction is currently misleading in making it seem like the virologic test within 4-6 weeks is all that encompasses EID. This comment is noted. 

Response - The introduction has been revised to include the EID cascade to clarify that Authors are interested in facilitators and barrier both to initiating EID and completion once started.

Comment 7 - I don’t see the checklist but perhaps the editor does. In any case, it does look like the authors incorporated much of these criteria. Thank you. 

Response - The checklist was included during the earlier submission. We have however included it to the attachments.

Comment 8 -I appreciate the authors saying they used “criterion sampling” but it would be beneficial to understand what the criterion was they were sampling on (mothers with certain characteristics, experiences…?) Saying you used criterion sampling is good, but you need to go a step further and elaborate on the criteria applied. You also mention “eligibility criteria” so should define that somewhere as well. 

Response -The criteria utilised for the selection of HIV positive mothers and health workers has been included.

Comment 9 - The section looks better, but the quote is still a little difficult to comprehend (“so they seeking treatment…”). Can the syntax be improved? (perhaps “so [the idea of] seeking treatment…”) 

Response- The quote has been revised for clarity.

---

## [Decision Letter · Decision Letter 2]

28 Jan 2021

Factors influencing the delivery and uptake of early infant diagnosis of HIV services in Greater Accra, Ghana: a qualitative study

PONE-D-20-16385R2

Dear Dr. Ankah,

We’re pleased to inform you that your manuscript has been judged scientifically suitable for publication and will be formally accepted for publication once it meets all outstanding technical requirements.

Kind regards,

Claudia Marotta

Academic Editor

PLOS ONE

Additional Editor Comments (optional):

dear authors congratulations

Reviewers' comments:

Reviewer's Responses to Questions

**Comments to the Author**

1. If the authors have adequately addressed your comments raised in a previous round of review and you feel that this manuscript is now acceptable for publication, you may indicate that here to bypass the “Comments to the Author” section, enter your conflict of interest statement in the “Confidential to Editor” section, and submit your "Accept" recommendation.

Reviewer #1: All comments have been addressed

2. Is the manuscript technically sound, and do the data support the conclusions?

Reviewer #1: Yes

3. Has the statistical analysis been performed appropriately and rigorously? 

Reviewer #1: N/A

4. Have the authors made all data underlying the findings in their manuscript fully available?

Reviewer #1: Yes

5. Is the manuscript presented in an intelligible fashion and written in standard English?

Reviewer #1: Yes

6. Review Comments to the Author

Reviewer #1: Thank you for addressing my comments adequately. I believe this manuscript is now suitable for publication.

7. PLOS authors have the option to publish the peer review history of their article (what does this mean?). If published, this will include your full peer review and any attached files.

Reviewer #1: No

---

## [Editor Report · Acceptance letter]

3 Feb 2021

PONE-D-20-16385R2 

Factors influencing the delivery and uptake of early infant diagnosis of HIV services in Greater Accra, Ghana:  a qualitative study 

Dear Dr. Ankrah:

I'm pleased to inform you that your manuscript has been deemed suitable for publication in PLOS ONE. Congratulations! Your manuscript is now with our production department. 

Kind regards, 

on behalf of

Dr. Claudia Marotta 

Academic Editor

PLOS ONE